# The Relationship between Occupational Fatigue and Well-Being: The Moderating Effect of Unhealthy Eating Behaviour

**DOI:** 10.3390/bs14010032

**Published:** 2024-01-02

**Authors:** Tingyu Wu, Xiaotong Tan, Yuying Li, Yongqi Liang, Jialin Fan

**Affiliations:** 1School of Psychology, Shenzhen University, Shenzhen 518000, China2021125852@email.szu.edu.cn (Y.L.);; 2The Shenzhen Humanities & Social Sciences Key Research Bases of the Center for Mental Health, Shenzhen 518060, China

**Keywords:** eating behaviour, occupational fatigue, well-being, work stress

## Abstract

Tech giants are large, well-known internet and technology companies. Employees of such companies are generally expected to work fast and for long periods of time, causing them to experience high occupational fatigue. The relationship between occupational fatigue and well-being is complex. Furthermore, in the context of the workplace, unhealthy eating behaviour may be used as a mechanism to cope with fatigue and stress. This study explored the relationship between occupational fatigue, well-being and unhealthy eating behaviour within this specific professional population. Study 1 used qualitative research methods, in which in-depth interviews were conducted with staff working at 13 tech giants in Shenzhen, China (*N* = 50). The findings revealed that work-related stress and occupational fatigue are common among employees working for tech giants. Additionally, factors such as unhealthy eating behaviour, workload, working hours and workplace interpersonal relationships were found to influence occupational well-being. Study 2 involved a cross-sessional survey of 237 employees of tech giants. The results indicated that occupational fatigue negatively impacts occupational well-being and that unhealthy eating behaviours play a moderating role between occupational fatigue and occupational well-being. These findings highlight the significance of adopting appropriate measures to improve the situation and cope with the effects of occupational fatigue by managing unhealthy eating behaviours.

## 1. Introduction

Occupational fatigue is very common among employees who work highly intense jobs requiring long hours, especially employees of internet companies who engage in prolonged sedentary behaviour and screen exposure [1]. These conditions have various adverse consequences, such as cognitive decline, decreased performance, cervical spondylosis, sleep disturbances, decreased job satisfaction and even unhealthy eating behaviour [2,3,4]. Food cues include word, visual, taste and olfactory inputs that are connected to eating [5]. And various food cues can either encourage or discourage eating [6]. Occupational fatigue may cause individuals to avoid having to think, making it difficult to sustain targeted behaviours and resist food cues [7]. A study reported that fatigue and workload were positively related to emotional eating, uncontrolled eating and an increased percentage of calories from fat among working adults [8]. And an excessive number of food cues can promote overeating. Tech giants are famous for their afternoon tea culture. For instance, Facebook offices offer a wide variety of snacks and beverages, regularly replenishing their inventory to ensure an ample supply of desirable options [9]. When company perks include an abundance of meals and snacks, the likelihood of succumbing to food cues and uncontrolled eating behaviours among employees increases. The issue of unhealthy eating behaviour, such as eating during work or binge eating, may be especially significant for tech giants that boast outstanding staff benefits. Furthermore, unhealthy eating behaviour is an issue of occupational health and could potentially be associated with occupational fatigue.

Occupational well-being refers to the subjective evaluation of employee job satisfaction and personal work experiences in the workplace. Much research has been conducted on well-being at work, and related factors have also been widely discussed. Studies have developed models that consider general mental ability and factors such as occupation, health and subjective well-being [10]. A study on nurses demonstrated that job benefits, job stability and social biases against the profession were associated with different levels of well-being at work [11]. Another study on teachers using the Job Demand-Resources (JD-R) model, which proposes that the demands and resources in the work environment will have an impact on the performance of employees, concluded that professional, individual, organisational and social factors significantly impact teachers’ professional well-being [12]. Moreover, individuals’ positive views and responses to work have positive effects on well-being, while work stress has negative effects on well-being [13]. Therefore, occupational well-being has become a hot topic in the field of occupational health.

Occupational fatigue is defined as the physical and mental decline experienced by workers as a result of their jobs and can occur before, during and after work. It is associated with poor working conditions, cognitive decline [14] and various diseases [15,16]. According to the JD-R model, high-intensity work can lead to physical and mental discomfort, resulting in work stress that impacts occupational fatigue and well-being [17,18]. Furthermore, occupational fatigue can be explained by personal characteristics and the stability of personal emotions [19]. Additionally, one report revealed a positive correlation between workload and occupational fatigue, while teamwork was negatively correlated with occupational fatigue, meaning that occupational fatigue is negatively associated with worker health and safety [20]. However, the relationship between occupational fatigue and individual characteristics, such as unhealthy eating behaviours, remains unclear. 

In the field of occupational health, research has examined the factors that influence the occupational health of employees from a group perspective [21]. However, there is a dearth of studies on the occupational health problems experienced by the employees of tech giants. What is more, these workers play a vital role in the development of these companies and even their local economies, and work fatigue and an unhealthy state are bound to have a negative impact on work efficiency [22]. The work projects for which they are accountable undergo frequent changes and have brief cycles. The impacts of occupational fatigue, such as on mental and physical health, need to be further elucidated, as they may be connected to occupational well-being. A study showed that mental exhaustion can result in occupational fatigue, which can be detrimental to mental health [23]. Occupational fatigue caused by overtime and prolonged working hours increases the risk of occupational injuries and illnesses, such as cardiovascular disease and musculoskeletal disorders [24,25,26]. In addition, occupational health is related to individuals’ perceptions of well-being. Studies have shown that from the perspective of the impact of resource management strategies on occupational health, micro-breaks can reduce fatigue and enhance occupational well-being [27]. Therefore, the relationship between occupational fatigue and well-being is worth exploring.

Unhealthy eating behaviour refers to the consumption of excessive high-calorie foods, such as those high in sugar, resulting in uncontrollable overeating, which greatly increases the risk of obesity. To better understand different eating behaviours, the Three-Factor Eating Questionnaire (TFEQ) categorises them according to three dimensions: cognitive restraint, uncontrolled eating and emotional eating [28,29]. The questionnaire is a well-established tool for assessing eating behaviour. Previous studies have used TFEQ to explore the relationship between unhealthy eating behaviours and cell phone dependence [30]. Furthermore, negative emotions have an impact on eating behaviour [31,32,33]. Professionals are constantly under pressure at work, and those who are unable to effectively manage this pressure may engage in behaviours such as emotional eating [34]. 

Many researchers have studied these unhealthy eating behaviours. Malnutrition, obesity, increased risk of cardiovascular disease and even psychological distress can all be the result of unhealthy eating behaviours [35,36]. Unhealthy eating behaviours have become a globally concerning issue, occurring among various age groups, demographics and countries [37,38,39]. It is worth noting that investigations into unhealthy eating behaviours have covered a wide range of participants, including adolescents, depressed patients and college students [40,41,42,43]. Emotional eating among adolescents in particular is triggered and intensified by negative emotions [44]. However, there has been a paucity of research on unhealthy eating behaviours in the workplace, specifically among employees of tech giants. Therefore, this study aims to examine the role of unhealthy eating behaviour among this professional population, shedding light on this underexplored area.

The Demands-Resources and Individual Effects (DRIVE) model, one of the models used in the field of occupational fatigue, informs the research framework [45,46]. The DRIVE model includes work and personal characteristics that may influence well-being and outcomes. According to the DRIVE model, individual differences (e.g., coping styles, personality and health-related behaviours) play a significant role in fatigue. Specifically, the model suggests that individual differences would moderate the relationship between perceived stress and outcomes. Thus, this model demonstrates not only the effects of job demands and resources but also the impact of individual differences on work fatigue and health outcomes. Notably, the existing literature has not extensively examined the role of unhealthy eating behaviour as a personal characteristic in the context of the DRIVE model. Eating behaviour traits are characteristics of individuals that influence behaviour and do not change on a daily basis [47]. Studies have explored the moderating role of eating behaviour traits in the relationship between behaviour and health outcomes [48,49]. Furthermore, the role of eating behaviours in the workplace is also a topic of debate. Some researchers have argued that food constitutes a crucial work resource that can enhance employee health, leading to increased work efficiency [50]. However, some studies have demonstrated that negative emotions, such as anxiety, can contribute to unhealthy eating behaviours, including disordered and emotional eating [51]. It can be challenging for professionals to avoid anxiety in the workplace, especially in a high-pressure, fast-paced environment. Therefore, the current study proposes that unhealthy eating behaviour, as an individual characteristic, may be an important moderator of the relationship between fatigue and well-being.

In this study, we aim to explore the relationship between occupational fatigue, well-being and unhealthy eating behaviour. To the best of our knowledge, there is limited research focusing on this topic; therefore, this study makes a significant contribution to the current knowledge in the field of occupational health. We hypothesise that occupational fatigue reduces individual well-being at work and that unhealthy eating behaviour as a personal characteristic plays a moderating role in the relationship between occupational fatigue and well-being at work.

The study comprised two stages: Study 1 involved semi-structured telephone interviews with 50 employees from tech giants that each lasted approximately 40 min, focusing on their work-related experiences and eating behaviours. A qualitative analysis of the interview content was conducted to identify key themes and provide support for the selection of questions in the follow-up (Study 2) questionnaire. Study 2 featured a revised questionnaire according to the interview results of Study 1, and employees with more than half a year of experience in different industries were surveyed. Finally, the survey results were further analysed to determine the relationship between occupational fatigue, unhealthy eating behaviour and well-being, and a model based on the DRIVE model was constructed to provide implications for employees, enterprises and governments.

## 2. Study 1

### 2.1. Background and Purpose

This study selected employees who had worked for a tech giant for more than six months. Participants were interviewed about their work situations, daily lives, eating behaviours and well-being to address the issue of overtime and high workloads among employees in the tech industry. The policy background surrounding occupational health protection was also discussed.

### 2.2. Materials and Methods

The interview content was developed based on the DRIVE model and related research on occupational fatigue. Specifically, the characteristics of food consumption occasions, frequency, motivations and feelings before and after consumption were considered. Finally, a semi-structured interview outline was developed. The interviews were conducted in Chinese and did not involve any translation, but the verbatim transcripts below have been translated into English. The structure of the interviews was as follows: A total of 25 questions were asked, with 6 about basic information about careers, 6 about lifestyle, 8 about stress and managing stress at work and 5 about well-being. The interviews were carried out through one-on-one telephone conversations or online conference communication, which lasted an average of 40 min.

To ensure the authenticity and usefulness of the interviews, the interviewees were recommended by the researcher’s acquaintances. Additionally, to prevent tendentious answers, the research purpose of the interview was not disclosed to the interviewees before the interview, and some questions were used to mitigate the interviewees’ feelings of defensiveness. Before the interview, the participants’ rights were explained to them, and privacy was guaranteed. Participants had the right to terminate the interview at any time. The opportunity to cease interview cooperation and delete interview data was also made available to the participants after the interviews.

#### 2.2.1. Participants

The case study involved 50 employees of tech giants and conducted 50 separate interviews with them. The following criteria were used to choose the interview participants: work experience of 6 months or more, non-psychology major and they worked for an internet company that is sizeable, well-known and offers high salaries as a standard like Huawei.

The study contacted an initial group of interviewees working at tech giant companies through alumni networks. Then, this initial group of interviewees recommended colleagues and friends from tech giant companies to participate in the interviews. To ensure the identity of each interviewee, we verified that each interviewee’s employee status could be found in the corresponding company’s collaboration system.

The specific companies chosen were carefully selected and had to meet the following criteria: (1) The company mandates lengthy overtime that goes beyond the eight-hour workday; (2) the company’s employees are overworked, as evidenced by the fact that the tasks they are accountable for cannot be accomplished within the eight-hour workday, therefore requiring overtime or additional manpower; (3) the company has been listed, is a listed company or has been identified as a top enterprise by reputable investment institutions, and its market value ranks among the top in the science and technology industry; (4) the average salary of employees is higher than that of the majority of comparable companies; (5) the company has a certain reputation; and (6) the business operations of the enterprise are fast, and the business iteration cycle is fast.

First-line business positions (programmer, product manager, operations, marketing, etc.) and secondary non-functional positions (human resources, finance, legal, etc.) made up the majority of the surveyed positions in Study 1. Detailed demographic information is shown in Table 1.

#### 2.2.2. Analysis Tool

NVivo 11.0 was used by this study for the qualitative analysis. Grounded theory is an important qualitative analysis approach. NVivo is software that was developed based on this theory to support qualitative and hybrid searches, which is convenient for organising and analysing unstructured and qualitative data [52,53]. The first step was to compile the material from the 50 interviews into a verbatim transcript, which was then loaded into the software to reformat, add or remove data. Second, in the coding stage, the content from the initial data was extracted in accordance with the interview topic, classified and summarised. Then, the content was coded in accordance with the nodes and content logic. Finally, free node analysis and tree node analysis were used to combine the research findings.

### 2.3. Results

The 50 interviewees ranged in age from 20 to 39, of which 36% were male (*n* = 13) and 64% were female (*n* = 37). They were mainly engaged in non-technical positions in the internet companies. In terms of education, 80% of the interviewees had a bachelor’s degree. Furthermore, 90% were between 20 and 30 years old, and more than 70% had fewer than 3 years of work experience.

The research findings can be categorised into the following domains when integrated with the DRIVE model: basic information about work stress, basic information about occupational fatigue, individual differences in occupational fatigue, unhealthy eating behaviour and well-being. Study 1 served as an initial attempt to synthesise the primary findings, establishing a foundation for future inquiries based on the outcomes derived from the semi-structured interviews. In addition, the selected participants were involved in diverse job positions and working conditions, thus rendering them representative of the target population and supporting the possibility of generalising the research findings based on available population data and previous empirical evidence.

#### 2.3.1. Basic Information on Occupational Fatigue

According to the DRIVE model, occupational fatigue is a consequence of either excessive job demands or a lack of job resources [45,46]. In the interviews, participants attributed their fatigue primarily to work-related factors and interpersonal relationships (Figure 1). Specifically, 40 mentioned that job characteristics such as heavy workloads, long working hours and high work intensity contributed to their fatigue. However, 25 identified interpersonal relationships in the workplace as a prominent source of fatigue, because relationships, whether with superiors or colleagues, require effort. And the atmosphere of the team will also affect feelings in the workplace.

#### 2.3.2. Individual Differences and Occupational Fatigue

In the DRIVE model, individual differences are regarded as independent variables and subjective feelings, and well-being and other health-related indicators are regarded as dependent variables, suggesting that there is a significant correlation between them. The interviews focused on eating habits, exercise habits, sleep habits, stress coping styles and other aspects to explore whether individual differences affect food consumption and work feelings (Figure 2). Among the 50 interviewees, most led healthy lifestyles; 39 did not smoke, 39 ate regularly, 34 considered themselves to have a positive stress coping style, 26 exercised and 21 had good sleep quality.

**Figure 2 behavsci-14-00032-f002:**
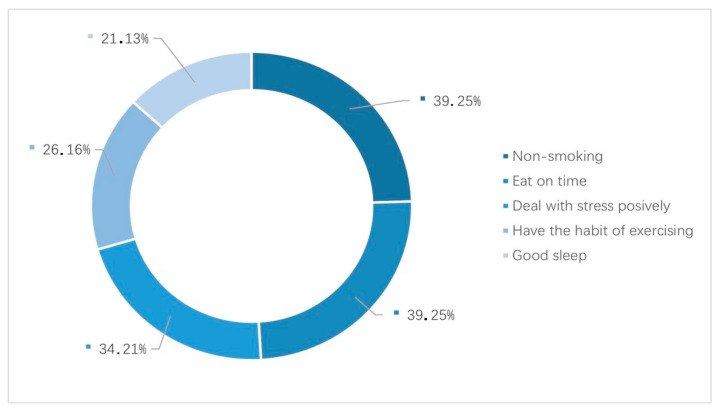
Individual differences in lifestyles.

However, the respondents also mentioned that work pressure, workload, workplace relationships and other factors affect their lifestyles. For example, they engaged in less exercise and suffered from decreased sleep quality:

‘Go to bed later and later… The lifestyle may not be very healthy’ (Subject 19, female, 30 years old, operations).‘Sometimes wake up earlier because of stress’ (Subject 22, female, 23 years old, operations).‘I had a breast operation in January this year due to excessive mental stress, which is now controlled by medication’ (Subject 14, female, 32 years old, operations).‘There is always stress; it must be because of work … I feel worse because I am so busy and have less time to exercise. I’m also mentally anxious because I am surrounded by excellent workers’ (Subject 11, female, 25 years old, researcher).

#### 2.3.3. Unhealthy Eating Behaviours

In this study, unhealthy eating behaviours were mainly assessed based on the ideas of food waste and healthy eating (Figure 3). In terms of food waste, 31 out of 50 respondents thought that they wasted food. Food waste behaviours include not being able to finish the food ordered, throwing it away when it expires, hoarding more food than you need and causing waste due to banquets or parties. The main food waste behaviour mentioned was that interviewees were unable to consume all the food provided in canteens, through takeaway and cooking. The semi-structured interview results showed that most of the employees’ inability to consume the food provided was not due to their subjective intentions but due to work pressure. Some participants shared the following:

‘If you are stressed, you will lose your appetite.’ (Subject 30, female, 25 years old, operations).‘In fact, I ate quite a lot. However, after coming to this company, the pressure caused me to lose my appetite.’ (Subject 25, female, 31 years old, marketing).

Regarding stress, negative emotions and adverse eating behaviours, 11 interviewees believed that their emotional eating behaviour occurred as a result of work pressure, and these 11 interviewees showed slightly excited emotions about emotional eating content. Eating more is one way to relieve work pressure. In the modern day, food consumption no longer occurs out of physiological needs but out of a psychological need to relieve pressure and obtain pleasure. One subject shared the following: ‘I have a lot of emotional eating. I tend to overeat when I’m stressed’ (Subject 33, male, 22 years old, community operations).

Another volunteer shared, ‘There is definitely emotional eating. If I work late or I’m stressed, I’m going to eat, even if it is very late. I also want to have some time alone and vent’ (Subject 44, male, 39 years old, operation supervisors). Another shared that ‘When I’m in a bad mood or I’m tired, I tend to overeat’ (Subject 28, female, 25 years old, operation).

**Figure 3 behavsci-14-00032-f003:**
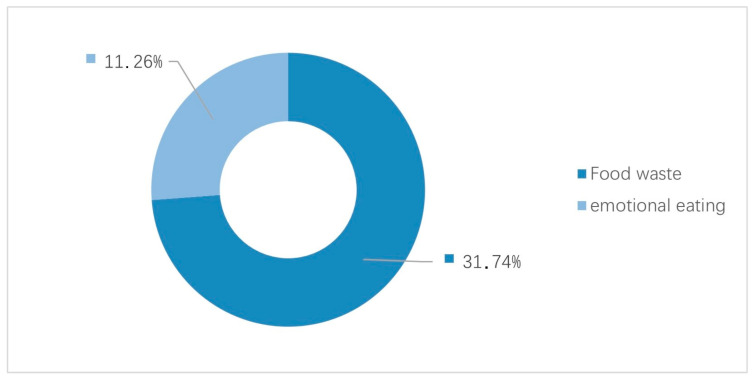
Unhealthy eating behaviour.

To enhance the findings on food waste behaviour, which were discussed in the interview, further cross-analysis (Table 2) was conducted, and the relationship between work stress and ‘clean your plate’ was found.

In total, 46 interviewees considered themselves to be under a lot of work pressure. Crossover frequency analysis revealed that among the 31 interviewees who engaged in food waste behaviour, 30 were under a lot of work pressure; that is, 96.8% of food waste behaviour resulted from work pressure.

The causes of food waste are shown in Table 3. Reasons included personal ones, such as picky eating or having small appetites; social and cultural reasons, such as banquet or dinner waste; and commercial reasons, such as waste due to promotion consumption or surplus due to large packaging volume. Furthermore, each of these causes was further divided into waste caused by portion size or waste caused by the evaluation of food (such as food taste, quality and variety). As can be seen from the table below, personal reasons were the key to individual food waste. Waste caused by external factors, such as social and cultural factors and commercial factors, was equally significant.

#### 2.3.4. Well-Being

Even in the face of high work pressure, positive feedback can make employees feel happy. As shown in Figure 4, 47 of the 50 interviewees mentioned well-being at work, and 28 believed that work itself can enhance well-being, such as through a sense of accomplishment in completing work, a high sense of value in work, admiration from superiors or the realisation of personal interests:

‘The point of well-being is to do something with a purpose’ (Subject 32, female, 24 years old, store management).‘When I solve a problem presented to me, I feel a sense of accomplishment and well-being’ (Subject 18, female, 24 years old, community operations).

Therefore, although work stress and fatigue are unavoidable, they do not necessarily bring unhappiness. Work can enhance employee well-being, highlighting the need to further elucidate the relationship between occupational fatigue and well-being.

Through the analysis of the interview results, two main types of user portraits are summarised:

Case 1: Mr. Chen (pseudonym; male, 39 years old, married with children and has old people in his family) works as a middle manager in a first-line internet company with 20 subordinates. Due to the distance of his residence, he needs to get up at six o’clock to commute to work at eight or nine o’clock. When his work ends is not fixed. Furthermore, the company is in the e-commerce industry, so working hours are long and the pace is very fast. Every major promotion in the e-commerce industry is accompanied by key performance indicators. As a middle manager, he is responsible for the business and the team. These working conditions have resulted in great stress and fatigue, causing him to become overweight. He also started to smoke and drink frequently. He usually eats takeout and can usually finish it, so he believes that his diet tends to be balanced. Mr. Chen believes that he has a positive stress coping style and optimistic outlook on life, even if he sometimes uses tobacco, alcohol and emotional eating to relieve occupational fatigue while shouldering the burden of caring for his family. However, he believes that well-being is not a given. Mr. Chen thinks his lifestyle is ‘pretty good’, aside from the occasional use of alcohol and tobacco. His feeling of well-being is enhanced when his family is harmonious and work achievements are recognised.

**Figure 4 behavsci-14-00032-f004:**
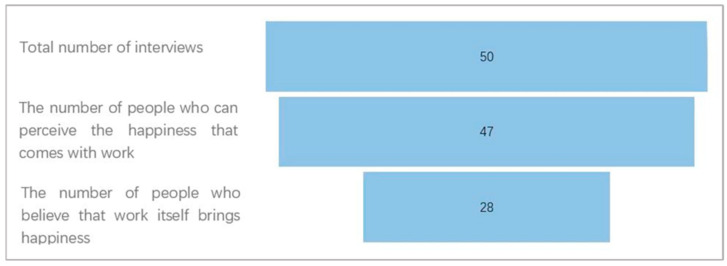
Well-being at work.

Case 2: Miss Fang (pseudonym; female, 31 years old, unmarried and childless) works as a sales executive in a first-line internet company. She believes that her working hours are not fixed and that there are many temporary tasks that are challenging. Her emotional overdraft is the main reason for her fatigue. She relieves the pressure by listening to music and through other ways. In terms of food consumption, the many meetings and the fast pace of her job caused her to start eating at irregular intervals and even skip dinner sometimes. Miss Fang herself has a big appetite, but occupational fatigue makes her eat less. She thinks it is unhealthy to order takeout every day. For these reasons, Miss Fang cooks herself on her own time, and she considers having meals with friends to be a ‘healing’ thing; for her, such activities are a way to relieve stress, which makes her happy.

### 2.4. Summary

The interview results showed that work stress and fatigue are common among the employees of tech giants. In addition to the characteristics of the work itself, other implicit factors are worthy of further investigation. Heavy workload, long working hours and fast working pace are relatively obvious factors, but the maintenance of workplace interpersonal relationships, recognition from leaders regarding work value, job accomplishment and other aspects need to be further studied. Moreover, employees’ eating behaviours also vary from person to person, and they reported that their appetites can be affected by psychological factors. Stress and negative emotions can reduce appetite or cause emotional eating. In addition, individual characteristics also affect employees’ judgment and feelings about work. Different stress coping styles generate differences in individual well-being. The above findings are consistent with the directions indicated by the DRIVE model, but the specific mechanisms of influence require further study.

## 3. Study 2

### 3.1. Links between Studies 1 and 2

Study 1 utilised a qualitative research method and conducted in-depth interviews with 50 employees of tech giants. The results showed that work pressure and occupational fatigue were prevalent among staff, and unhealthy eating behaviours mostly occurred in the context of work pressure. The DRIVE model emphasises the importance of individual differences in the process of attaining well-being. Thus, Study 2 was a cross-sectional study aimed at further exploring the impact of occupational fatigue on well-being at work and examining the moderating role of unhealthy eating behaviours.

### 3.2. Methods

#### 3.2.1. Participants and Procedure

Participants were employees of tech giants in China (*N* = 237), with a mean age of 26.43 years (standard deviation [SD] = 6.54), of which 54.85% were female (*N* = 130). The inclusion criteria were a minimum of six months of work experience and current employment. Of the participants, 77.64% had work experience of fewer than five years, which may be attributed to the fact that the technology industry is an emerging field in China.

An online survey was conducted, and snowball sampling was used to recruit participants. Participants were required to provide informed consent and had the right to withdraw from the study at any time. The School of Medicine Ethical Committee at Shenzhen University reviewed and approved this study.

In Study 2, after the initial formulation of the questionnaire outline, 30 employees of large factories were selected to participate in the questionnaire survey anonymously, and the number of questions was optimised. After the overall content was confirmed, the anonymous questionnaire survey was mainly published in the following ways: By reaching the social networks of some full-time alumni and employees of large factories, more similar groups were leveraged to participate in the survey; in particular, the grassroots managers within some large factories expressed their support after receiving the questionnaire survey invitation and carried out a certain degree of promotion within the company; the Intra-University communication community was also utilised. For example, alumni groups and job search experience exchange groups were utilised, as the job search goals and central topics of such communities are all related to technology companies. Based on the above methods, the researchers recovered certain data and carried out analysis work.

#### 3.2.2. Materials

The survey included questions about demographics, the Smith Happiness Questionnaire (SWELL) and the TFEQ-R21. Demographic information collected included gender, age, education and years of work.

Occupational fatigue and well-being at work were evaluated using the SWELL [54], which is based on the DRIVE model [46]. This questionnaire has been used to assess a variety of occupational groups, enabling the identification of general occupational risks [55,56]. This questionnaire was translated from English into Chinese using both forward and back translation and proved to have good reliability and validity in the Chinese occupational population [57]. The SWELL consists of 26 single-item questions, most of which call for answers on a 10-point Likert scale. Such single-item measures were valid and reliable [4,55,58]. In the current study, occupational fatigue and well-being at work were the main variables of interest (e.g., how physically, mentally or emotionally fatigued do you feel at work in general? Are you happy at work?). Additionally, work characteristics (workload and stress at work) were also considered and evaluated to establish associations with the main variables.

The TFEQ is one of the most widely used self-reported questionnaires. It is used in studies of the eating behaviours of normal weight and overweight individuals [59]. The original version contains 51 items, and a short version with 21 items was subsequently developed [60]. The shortened form is also more practical for use in epidemiological and clinical trials. The TFEQ-R21 has been proven to have commendable internal consistency and a stable factor structure across multiple studies [32,33,61]. In the current study, the Chinese version of the TFEQ-R21 was used with authorisation by the scale’s developer, Dr Karlsson. The questionnaire was composed of 21 items pertaining to three dimensions: uncontrolled eating, cognitive restraint and emotional eating. The study investigated the moderating effect of unhealthy eating behaviour on the relationship between fatigue and well-being. To assess overall unhealthy eating behaviour across all dimensions, the total mean score derived from the TFEQ-R21 was chosen as the primary indicator. Higher scores on the scale are indicative of a greater inclination towards unhealthy eating behaviour.

#### 3.2.3. Analyses

Data analysis was conducted using SPSS 25 and PROCESS Macro version 4.2. Descriptive analyses were used to determine the participants’ scores for each variable. Pearson’s correlation was calculated to assess the relationships between the variables. Moderation effect tests were then conducted to examine the impact of occupational fatigue on well-being at work and to identify the moderating role of unhealthy eating behaviours. Before conducting the moderation analysis, both the independent and moderating variables were centred to eliminate the issue of multicollinearity.

### 3.3. Results

#### 3.3.1. Descriptive Statistics and Correlations

The mean (M), SD and correlations of the study variables are shown in Table 1. Participants had a mean occupational fatigue score of 6.66 ± 1.89, a mean workload score of 6.40 ± 1.97 and a mean stress score of 6.57 ± 1.87, indicating that fatigue and stress problems were prevalent among the participants. Pearson correlation was conducted as shown in Table 4. Occupational fatigue was significantly positively correlated with work characteristics, including workload (r = 0.506, *p* < 0.01) and stress at work (r = 0.714, *p* < 0.01). There was a negative relationship between occupational fatigue and well-being at work, indicating that high levels of fatigue are associated with poor well-being. In addition, no significant association was found between either occupational fatigue or well-being at work and unhealthy eating behaviours, underscoring the potential moderating effects of unhealthy eating behaviours.

#### 3.3.2. Moderating Effect of Unhealthy Eating Behaviours

To investigate the moderating effect of unhealthy eating behaviours on the association between occupational fatigue and well-being at work, the procedure of PROCESS Macro model 1 for SPSS proposed by Hayes was used [62]. A bootstrap with 5000 samples and a 95% confidence interval was utilised for verification. Mean centred the independent variable and the moderator variables. And the gender variable was controlled.

As shown in Table 5, occupational fatigue was found to have a significant negative effect on well-being at work (β = −0.149, *p* < 0.05). Moreover, the interaction term between occupational fatigue and unhealthy eating behaviours (TFEQ-R21) showed a significant positive effect on well-being at work (β = 0.337, *p* < 0.01), and the confidence interval for bootstrapping [0.097, 0.577] did not include zero. Thus, it can be suggested that unhealthy eating behaviours moderate the relationship between occupational fatigue and well-being at work.

To further understand the moderating effect, the effect of occupational fatigue on well-being at work was analysed based on different levels of unhealthy eating behaviours. Unhealthy eating behaviours were divided into high-score (M + SD) and low-score (M − SD) groups. In Figure 5, the slope reflects the predictive effect of occupational fatigue on well-being at work. As unhealthy eating behaviours increase, the predictive effect of occupational fatigue on well-being at work becomes progressively insignificant (low-score group: simple slope = −0.311, t = −3.894, *p* < 0.01; high-score: simple slope = 0.013, t = 0.150, *p* = 0.881).

More specifically, Johnson–Neyman analysis was conducted to identify the significant region of the conditional effect. The threshold value for the moderator that defined the Johnson–Neyman significance region was determined to be 2.502 (t = −1.970, *p* = 0.05), indicating that the conditional effect of occupational fatigue was significant when the TFEQ-R21 score was lower than 2.502.

### 3.4. Summary

This cross-sectional study explored the relationship between occupational fatigue and well-being at work and examined the moderating role of unhealthy eating behaviours. The results revealed a significant positive correlation between occupational fatigue, workload and work stress. Higher levels of occupational fatigue were associated with lower well-being at work. However, no significant associations were found between occupational fatigue or well-being at work and unhealthy eating behaviours, suggesting a potential moderating role of unhealthy eating behaviours. The predictive effect of occupational fatigue on well-being at work became progressively insignificant as unhealthy eating behaviours increased, and Johnson–Neyman analysis further demonstrated that the conditional effect of occupational fatigue was significant for participants with TFEQ-R21 scores below 3.835. These findings suggest that to some extent, unhealthy eating behaviours may partially alleviate the negative effects of occupational fatigue on well-being at work, but boundary conditions must be taken into account.

## 4. Discussion

This study investigated the relationship between occupational fatigue, unhealthy eating behaviour and well-being among tech giant employees. The findings indicate the prevalence of work stress and occupational fatigue among this professional population. Additionally, various factors, including unhealthy eating behaviours and high workloads, were found to influence occupational well-being. Moreover, occupational fatigue can negatively predict well-being at work, and an increase in unhealthy eating behaviour within the boundary can lead to a decrease in this prediction.

This study reinforces the findings from earlier studies that occupational fatigue has a detrimental impact on occupational well-being. However, further investigation is necessary to elucidate the factors that influence the relationship between occupational well-being and occupational fatigue. According to research, compared to nonowners, business owners have stronger perceived job control, which is associated with higher job satisfaction and lower emotional exhaustion [63]. Furthermore, cognitive emotional regulation—the ability of individuals to reevaluate thought processes—can help individuals increase their flexibility to cope with stress, thereby reducing emotional exhaustion and improving subjective well-being [64]. Similar to perceived job control and cognitive emotional regulation, there are a variety of individual differences that can impact how occupational fatigue and well-being are related.

Individual differences have an impact on how each person perceives their level of occupational fatigue and well-being. In the first study, the qualitative analysis revealed that individuals’ assessments of job-related stressors are multidimensional. People also reported varying levels of occupational fatigue as a result of differences in stress coping mechanisms, lifestyle choices and other factors. Despite the stress and fatigue of work, some people believed that work itself can increase their well-being. In addition, numerous previous studies have found that individual differences influence factors associated with occupational fatigue. For instance, the degree of match between people and organisations has an impact on job performance and job satisfaction, and tasks that are not appropriate for employees’ skill sets will result in job discontent [65]. Intrinsic work motivation also reduces the negative influence of learning opportunities on fatigue and enhances the beneficial effect of job autonomy on job engagement [66]. Thus, previous research has shown that individual differences influence occupational fatigue.

The present study further enriches this field. With the help of the DRIVE model, the findings have further proved the different effects of occupational fatigue in individuals with differences in unhealthy eating behaviours. Unhealthy eating behaviour plays a role not as a directly related variable but as a moderating variable.

Overall, researchers anticipate that people can decrease fatigue and enhance occupational well-being through psychological adjustment and objective condition intervention. Furthermore, employees must develop a healthy working and lifestyle based on their actual situations, taking into account their individual differences.

This study shows that unhealthy dietary behaviour can partially regulate the impact of occupational fatigue on well-being. Among individuals with low unhealthy eating behaviour, occupational fatigue leads to lower levels of well-being as unhealthy eating behaviour increases. However, for individuals with high levels of unhealthy eating behaviour, occupational fatigue does not significantly impact well-being. This suggests other factors may influence the relationship between occupational fatigue and well-being in the highly unhealthy eating behaviour group. However, even though high unhealthy eating behaviour does not predict the negative effects of fatigue on well-being, we still do not advocate for it because unhealthy eating habits have adverse effects on health. Research shows that the lower the intake of fruits and vegetables, the higher the stress levels [67]. Additionally, a diet low in omega-3 polyunsaturated fatty acids and high in fatty foods is also associated with depression [68].

This study broadened the range of research subjects. In contrast to earlier studies, this work studied the Chinese working populace, specifically the employees of tech giants who typically deal with high levels of pressure and fast-paced work. Previous studies focused more on fatigue and well-being within specific industries, such as education, medicine and manufacturing. Furthermore, this study goes beyond the conventional focus by investigating the role of unhealthy eating behaviour, a fundamental and critical factor, within the work environment. Therefore, it represents a significant extension of the existing research.

We believe this research has significant implications for individuals, companies and even policies in regulating occupational fatigue and unhealthy eating behaviours. From an individual’s point of view, our research can offer psychological and science-based explanations of employees’ dietary, work and psychological issues, assist more employees in confronting health issues and comprehending the psychological mechanisms underlying unhealthy eating behaviours and occupational fatigue and assist individuals in developing healthy work habits and lifestyles. From the perspective of companies, this study can serve as a guide for setting up more thorough and effective employee care policies and nutritional supplement measures. By examining how occupational fatigue affects employee well-being, enterprises may pay closer attention to the occupational health of their workforce and promote a healthy eating culture through policy changes. At the same time, based on an understanding of employees’ eating habits, companies can reduce unnecessary food expenses. From a policy perspective, this study explored the relationship between occupational fatigue and unhealthy eating behaviour and well-being, which can provide theoretical support for companies to develop healthy dietary programmes for professionals and promote the implementation of ‘China Healthy Lifestyle for All (2017–2025)’.

This study has the following limitations. First, most of the participants who completed the questionnaire were professionals with fewer than five years of work experience and a college degree. This could have led to a biased sample. Despite the fact that the participants came from various spheres of life, the sample age group was concentrated. Second, this study did not consider the link between the sub-concepts of fatigue and unhealthy eating behaviour. For instance, there are two types of fatigue: acute fatigue and chronic fatigue. Unhealthy eating behaviour also includes different dimensions, such as time point, quantity, the reinforcing value of food and meal context, such as watching TV while eating [69,70,71], and acute fatigue may lead to bad eating habits [72]. The underlying reasons for unhealthy eating behaviour can be further elucidated, as different aspects of unhealthy eating behaviour may be connected to a variety of dimensions of fatigue. Additionally, our measurement methods were all self-subjective reports, and there was a lack of more objective records or instruments to quantify the degree of unhealthy eating behaviour and occupational fatigue.

Extensions of this research are possible, taking into account the limitations of the present study. Even though subjective sensations have an impact on occupational fatigue, physiological changes provide effective quantitative evidence. For this reason, future studies could quantify the physical and physiological aspects of occupational fatigue. This study largely relied on the SWELL questionnaire to gauge employee fatigue; however, future research can include cortisol testing, heart rate measurements, breathing analysis and other techniques to measure physiological status. Future research can also separate occupational fatigue into acute fatigue and chronic fatigue, which should be researched in conjunction with unhealthy eating behaviour to examine the various short- and long-term implications of occupational fatigue. Furthermore, the model can potentially include other variables related to character attributes, such as a person’s work ethic and pressure tolerance.

## 5. Conclusions

The findings of this study indicate that work stress and occupational fatigue are prevalent among the employees of tech giants. In addition to the inherent characteristics of the work itself, individuals’ different stress coping styles can influence their levels of happiness at work. Furthermore, occupational fatigue has a negative impact on happiness at work. Importantly, unhealthy eating behaviours play a moderating role between occupational fatigue and happiness at work, and they can mitigate the effects of occupational fatigue on happiness at work within a certain range. The findings of this study make a significant contribution to the understanding of the occupational health of tech giants. Our research highlights the prevalence of these issues and the impact they have on well-being at work. Additionally, we identified an important factor that influences individuals’ levels of happiness at work—their unhealthy eating behaviour. This reminds us that while unhealthy eating behaviour can temporarily alleviate occupational fatigue, excessive indulgence does not alleviate the negative effects of occupational fatigue on job happiness. Consequently, companies can provide appropriate portions of food as a means to relieve employee stress and fatigue. However, relying solely on excessive unhealthy eating behaviour to relieve occupational fatigue during work is not recommended.

## Figures and Tables

**Figure 1 behavsci-14-00032-f001:**
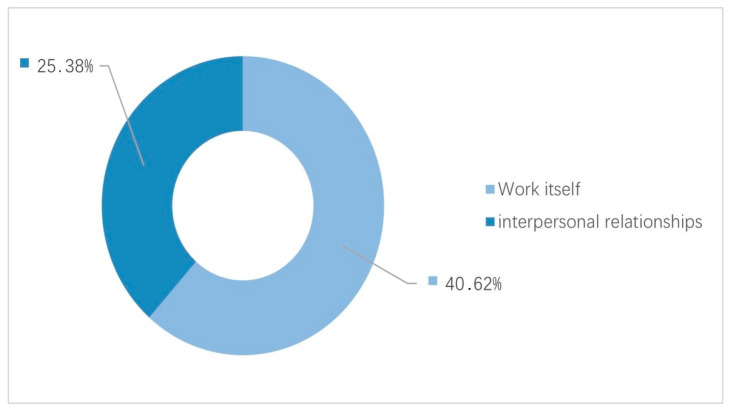
Sources of occupational fatigue. The left and right sides of the ordinates in Figure 1, respectively, indicate the number of people and the percentage of the total. The same is true for Figure 2, Figure 3 and Figure 4.

**Figure 5 behavsci-14-00032-f005:**
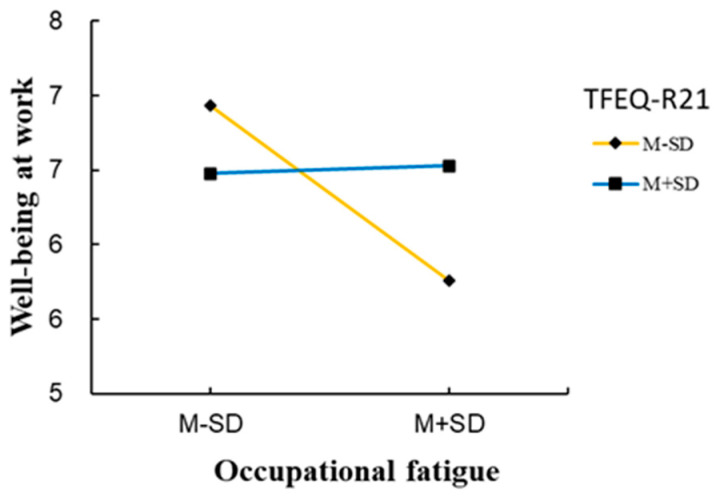
Moderating effect of unhealthy eating behaviours on the relationship between occupational fatigue and well-being at work.

**Table 1 behavsci-14-00032-t001:** Basic demographic information of interviewees.

Variable	Category	Frequency	Percentage (%)
Gender	Male	13	26
	Female	37	64
Age	20–30	45	90
	31–40	5	10
Education	Bachelor’s degree	40	80
	Postgraduate degree	9	18
	Non-university certificate or college or university certificate	1	2
Working experience	1 year or less	17	34
	1–3 years	19	38
	4–6 years	4	8
	7–9 years	5	10
	More than 9 years	5	10
Post	Operations	20	40
	Product	12	24
	Marketing	7	14
	Human resources	2	4
	Data analysis	1	2
	Market	1	2
	Business	1	2
	Channel	1	2
	Management	1	2
	User research	1	2
	Analyst	1	2
	Communication	1	2
	Consulting	1	2

**Table 2 behavsci-14-00032-t002:** Cross-analysis of work stress and the idea of ‘clean your plate’.

	Be Able to ‘Clean Your Plate’	Cannot ‘Clean Your Plate’	Occasionally Cannot ‘Clean Your Plate’
High working pressure	17	27	3
Low working pressure	2	1	0

**Table 3 behavsci-14-00032-t003:** Reasons for food waste.

	Commercial Reasons	Social and Cultural Reasons	Personal Reasons
The evaluation of food	21	23	59
Portion size	19	20	37

**Table 4 behavsci-14-00032-t004:** Means, standard deviations and correlations of the variables.

Variable	M ± SD	1	2	3	4	5	6	7	8
Occupational fatigue	6.66 ± 1.89	1							
Workload	6.40 ± 1.97	0.506 **	1						
Stress at work	6.57 ± 1.87	0.714 **	0.675 **	1					
Well-being at work	6.46 ± 1.78	−0.169 **	0.105	−0.010	1				
Unhealthy eating behaviours	2.42 ± 0.48	0.105	0.133 **	0.110	0.017	1			
Cognitive restraint	2.60 ± 0.55	0.006	0.093	0.077	0.117	0.369 **	1		
Uncontrolled eating	2.28 ± 0.59	0.106	0.086	0.067	−0.011	0.899 **	0.033	1	
Emotional eating	2.45 ± 0.77	0.105	0.126	0.109	−0.035	0.892 **	0.055	0.793 **	1

** *p* < 0.01.

**Table 5 behavsci-14-00032-t005:** Moderating effect of poor eating behaviours (TFEQ-R21).

Model	DV: Well-Being at Work							
	Coefficient	SE	t	LLCI	ULCI	F	R	R^2^
Gender	−0.264	0.229	−1.152	−0.714	0.187	4.140 **	0.258	0.067
Occupational fatigue	−0.149	0.060	−2.465 *	−0.268	−0.030			
Poor eating behaviours	0.166	0.239	0.694	−0.304	0.636			
Occupational fatigue × poor eating behaviours	0.337	0.122	2.767 **	0.097	0.577			

** *p* < 0.01 * *p* < 0.05. Abbreviations: DV = dependent variable; SE = standard error; LLCI and ULCI refer to lower and higher limits of 95% confidence interval.

## Data Availability

The datasets used and analysed in the current study are available from the corresponding author, J.F. (fanjl@szu.edu.cn), upon reasonable request.

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
