# Peer review of "The Relationship between Occupational Fatigue and Well-Being: The Moderating Effect of Unhealthy Eating Behaviour"

_behavsci, 2024, doi:10.3390/bs14010032_

Round 1

Reviewer 1 Report

Comments and Suggestions for Authors

Dear authors,

Here are the general considerations about the manuscript.

I don't understand the need to mention the names of the companies, called tech giants, in the study. It doesn't seem very appropriate.

It wasn't clear to me how the employees of the companies were approached for the interview. How were phone numbers obtained, for example?

On line 63, the JD-R model is suddenly mentioned, without offering any meaning for the acronym.

I don't think it's appropriate to use the expression "work happiness" (line 86) in research in the field or in occupational health literature, because it's vague and not very tangible. I think the aim is to analyze the occurrence of well-being or similar events.  

It seems to be an obvious conclusion about the effect of fatigue: occupational fatigue has a negative impact on happiness at work. I think that, given the structure of the study, the conclusion could have been better developed.

Check typo on line 153.

Lines 165 to 169: I understand that the information is more compatible with the results section.

Reviewer 2 Report

Comments and Suggestions for Authors

An interesting study. I have a few comments for the authors.

(i) Kindly state the aim and objectives of the study in the introduction, not just what is hypothesized. This should be stated before Line 124, which talks about the structure of the study.

(ii) State the study's contribution to knowledge in the introduction and conclusions sections.

Reviewer 3 Report

Comments and Suggestions for Authors

The present study focuses on a line of research of great interest for occupational health. Relating occupational fatigue and well-being to eating behaviors and other personality characteristics makes the topic highly relevant for the design of prevention programs in the type of companies that have been studied.

However, after reviewing this manuscript, there are conceptual and methodological issues that must be clarified since, from my point of view, they are serious limitations to this study.

1.     The study focuses on work fatigue and its relationship with well-being at work, highlighting the moderating role of poor nutrition. If these are the variables on which the research focuses, the introduction should focus on defining each of these variables and justify the importance of investigating the relationship between them. To do this, it is advisable to refer to previous and recent studies, making clear the novelty that this study provides. Regarding eating behaviors, why do they use the term poor eating behavior? Could you justify with previous studies that the use of this term is correct? In the scale used they talk about 3 dimensions. Is poor nutrition indicated on that scale as a global term?

2.     The authors point out that they start from the DRIVE model. However, the objective of their study does not evaluate personal variables such as those indicated in the model (for example, coping strategies). It could be understood that eating poorly at work is a maladaptive coping strategy for fatigue, if we focus on the emotional eating approach. However, this does not appear to be the approach from which the authors assume that poor eating at work moderates the relationship between fatigue and well-being. What theoretical approaches are they based on for their starting hypotheses? Couldn't we expect a mediating role instead of a moderator?

3.     Objectives for each study are not operationally defined. It is not clear what the first study contributes to the second. Nor is it understood why data on stress and work overload are provided if they are not part of the objective of the study and its starting hypothesis.

4.     In study 1 it is indicated that the interview is carried out by telephone. It would be convenient to describe the procedure for recruiting participants, how the identity of the participants was ensured. Were the conversations recorded? What security measures were used to protect the data?

5.     In study 2. A procedure section is missing detailing how the sampling was carried out, whether the survey was anonymous, how the tests were applied, etc.

6.      Regarding the evaluation instruments, the authors point out that they used The Smith Happiness Questionnaire (SWELL) to evaluate fatigue and well-being at work. Could you indicate with which specific items of the questionnaire you have evaluated each variable? On the other hand, the questionnaire is not validated in the population under study. The authors indicate that they have carried out the translation-back translation, however, it is not indicated that analyzes have been carried out to verify the reliability and validity of the test.

7.     Regarding the statistical analyses, it is indicated that correlations have been made but not of what type. The effect size has not been calculated either. The correlations between the variables fatigue, poor eating and well-being are not significant. One could consider carrying out the correlations by decomposing the poor eating variable into the three dimensions evaluated by the questionnaire used. On the other hand, if the correlations are not significant, it does not make much sense to propose inferential analyses. Regarding the moderation model, no data is provided on its significance.

8.     Has the sex variable been controlled?

9.     Regarding the discussion and conclusions, it is questionable whether statements and conclusions can be made based on the statistical analyzes that have been carried out.

Round 2

Reviewer 3 Report

Comments and Suggestions for Authors

I thank the authors for taking my suggestions into account. I think it is much clearer now and I have been able to resolve issues. I have detected that there were some errors due to the changes made. I recommend that you thoroughly review the final version.